# Novel Therapeutic Approaches with DNA Damage Response Inhibitors for Melanoma Treatment

**DOI:** 10.3390/cells11091466

**Published:** 2022-04-26

**Authors:** Luisa Maresca, Barbara Stecca, Laura Carrassa

**Affiliations:** 1Tumor Cell Biology Unit, Core Research Laboratory, Institute for Cancer Research and Prevention (ISPRO), Viale Gaetano Pieraccini 6, 50139 Florence, Italy; luisamaresca@hotmail.com; 2Fondazione Cesalpino, Arezzo Hospital, USL Toscana Sud-Est, Via Pietro Nenni 20, 52100 Arezzo, Italy

**Keywords:** DNA damage response, melanoma, PARP, ATM, CHK1, WEE1, ATR, inhibitors, combined therapy

## Abstract

Targeted therapies against components of the mitogen-activated protein kinase (MAPK) pathway and immunotherapies, which block immune checkpoints, have shown important clinical benefits in melanoma patients. However, most patients develop resistance, with consequent disease relapse. Therefore, there is a need to identify novel therapeutic approaches for patients who are resistant or do not respond to the current targeted and immune therapies. Melanoma is characterized by homologous recombination (HR) and DNA damage response (DDR) gene mutations and by high replicative stress, which increase the endogenous DNA damage, leading to the activation of DDR. In this review, we will discuss the current experimental evidence on how DDR can be exploited therapeutically in melanoma. Specifically, we will focus on PARP, ATM, CHK1, WEE1 and ATR inhibitors, for which preclinical data as single agents, taking advantage of synthetic lethal interactions, and in combination with chemo-targeted-immunotherapy, have been growing in melanoma, encouraging the ongoing clinical trials. The overviewed data are suggestive of considering DDR inhibitors as a valid therapeutic approach, which may positively impact the future of melanoma treatment.

## 1. Introduction

DNA damage occurs often in cells under the pressure of exogenous agents, such as exposure to ultraviolet (UV) light, ionizing radiation and chemicals, as well as endogenous factors (e.g., replication errors, oxidative stress), which eventually cause single strand breaks (SSB) or double strand DNA breaks (DSB) [1,2]. To cope with these events and to maintain genomic stability, cells activate the DNA damage response (DDR) signaling pathway [3,4] that detects and processes DNA damage. The ataxia-telangiectasia and Rad3-related (ATR) and ataxia-telangiectasia mutated (ATM) kinases are the two key upstream sensors of DNA damage [5]. ATM is activated in response to DNA damage, inducing DSB, while SSB, deriving from the process of DSB at resection and caused by replication stress, activates ATR [6,7,8]. ATR is recruited to the sites of DNA breaks by interacting with ATR-interacting protein (ATRIP), which recognizes RPA-coated ssDNA [9]. The main downstream target of ATR is the checkpoint kinase 1 (CHK1), which becomes activated by phosphorylation and, in turn, inactivates by phosphorylation the phosphatases CDC25A and CDC25C, respectively, involved in the dephosphorylation and activation of cyclin dependent kinases 2 (CDK2) and 1 (CDK1). Thus, the maintenance of CDKs in their phosphorylated and inactivated form blocks S and M phase entry [10,11,12]. In parallel, the tyrosine kinase WEE1, by negatively regulating by phosphorylation the CDKs activity in S and G2 phases, also plays a crucial role in controlling S and M phase entry [13,14]. The DDR pathway operates in cells to identify DNA damage and consequently to block cell cycle division, providing time for accurate DNA repair or promoting apoptosis, if the damage is unrepairable [2,4,15].

Several types of DNA repair systems, including non-homologous-end-joining (NHEJ), homologous recombination (HR), mismatch repair (MMR), nucleotide excision (NER) and base excision repair (BER), play an important role in maintaining cell viability and genomic stability [2,16,17]. The deregulation of the DDR pathway can cause mutations and genomic instability, thus, leading to cancer development [2,15,16,18]. Nevertheless, defects in the DDR possibly provide a therapeutic advantage, since cells with defective DDR signaling depend on compensatory pathways that can be exploited therapeutically. In the last two decades, a number of DDR inhibitors have been developed and evaluated in clinical settings, including PARP, ATR, ATM, CHK1 and WEE1 inhibitors [4,15,17,19].

In this review, we summarize preclinical and clinical studies on these novel DDR inhibitors in melanoma (Figure 1). We also discuss recent findings indicating that DDR inhibitors may be a strategy to strengthen the efficacy of current standard-of-care targeted therapy and immunotherapy in melanoma.

## 2. Melanoma

Melanoma is a skin cancer that develops from the malignant transformation of melanocytes, pigment producing cells which derive from the neural crest. The pigments of melanin defend against UV radiations, protecting cells from DNA damage [20]. Melanoma is the fifth most common cancer in men and the sixth in women. Melanoma accounts only for a small percentage of all skin cancers, but it causes the majority of skin cancer-related deaths [21]. Although melanoma is one of the few cancers with an increasing global incidence, paradoxically, the mortality rates appear to be rising less rapidly [22].

Genetic-landscape analyses revealed that cutaneous melanoma is the cancer with the highest mutational burden (>10 mutations per megabase) and is characterized by frequent C→T mutations, which are caused by the incorrect repair of UV-induced covalent bonds between adjacent pyrimidines [23,24]. The high mutation rate contributes to several aspects of melanoma genesis; however, certain alterations are considered driver, as they are involved in the transformation of melanocytes, and contribute to the onset and progression of this dismal disease.

The malignant transformation of melanocytes into melanoma occurs through a number of sequential events that lead to the constitutive activation of several oncogenic signaling pathways and inputs. A typical feature of benign nevi formation is the acquisition of the BRAF activating mutation (BRAFV600E). Point mutations in BRAF and NRAS are often mutually exclusive, with NRAS alterations sometimes already present in nevi [25]. The progression into melanoma in situ normally requires additional steps, such as alterations in the telomerase reverse-transcriptase (TERT) promoter [26]. The altered expression of telomerase makes melanoma cells replicative immortal [27]. After the accumulation of various mutations in the genes controlling cell cycle, such as cyclin-dependent kinase inhibitor 2A (CDKN2A), TP53, phosphatase and tensin homolog (PTEN) and genes encoding SWI/SWF chromatin remodeling complex subunits, particularly *ARID1A* and *ARID2*, primary melanoma enters the vertical growth phase and becomes malignant melanoma [28,29]. A high mutational burden and increased copy number alterations characterize the last steps of progression [29].

The Cancer Genome Atlas Network proposed a classification of cutaneous melanomas into four different genetic subtypes, based on the analysis of DNA, RNA and protein expression of more than 300 primary and metastatic melanomas (MM). These subtypes are represented by BRAF mutant melanomas, which account for approximately 50% of melanomas; NRAS-, KRAS-, and HRAS-mutant melanomas, which are about 25%; NF1-mutant melanomas (15% of melanomas); and triple-wild-type melanomas, which account for the remaining 10% of cases [30].

Despite several evidences pointing to a correlation between metastatic progression and increase in mutation burden, genomic instability and alterations in the DDR pathway in different types of cancer, little is known about the alterations and differential expression of DDR genes during melanoma progression. A previous study showed an increased expression of genes associated with DNA repair from primary to metastatic melanomas, suggesting that genetic stability might be required for melanoma cells to metastasize [31].

### Homologous Recombination and Melanoma

Homologous recombination (HR) is based on template-directed DNA repair synthesis in order to obtain an error-free repair of DSBs [32]. A deficiency in the HR pathway, such as *BRCA1* and *BRCA2* mutations, causes genomic instability and contributes to cancer development [33]. In the last few years, several studies have assessed the frequency of HR alterations in cutaneous melanoma. NGS data from Foundation Medicine (FMI) (Cambridge, Ca) in a large cohort of melanoma samples identified HR alterations in 33.5% of the cases. The analysis of 1088 melanoma patients in cBioPortal identified a high frequency (41%) of HR pathway mutations [34]. Another mutational screening in multiple solid tumors identified a 18.1% prevalence of HR alterations in melanoma samples [35]. In a recent study, Kim and colleagues reported that mutations in the HR-DDR pathway are frequent in cutaneous melanoma, showing that 21.4% of melanoma present alterations in at least one gene of the HR-DDR pathway. The most commonly mutated genes are *BRCA1*, *ARID1A*, *ATM*, *ATR* and *FANCA*. Interestingly, these alterations were significantly associated with thinner primary lesions, a higher fraction of head and neck primary melanoma, high tumor mutational burden (TMB) and a higher number of patients responding to anti-PD-1 therapy. However, they did not observe any significant correlation between HR genes status and overall survival [34]. Another work suggested that, in cutaneous melanoma, alterations in MMR and HR are associated with a high TMB, which is reported to be a predictive marker for cancer immunotherapy [36]. Two studies identified somatic *BAP1* (BRCA1-Associated Protein 1) mutations in approximately 5% of cutaneous melanoma and in 30–40% of primary uveal melanoma [37,38]. In uveal melanoma samples, these alterations were associated with a higher metastatic risk in comparison to wild type tumors and those with germline *BAP1* mutations [38].

Many HR genes (*BRCA1/2* and *BAP1*) are also related to syndromes that enhance genetic predisposition to cancer, including melanoma [39,40]. Germline mutations in *BAP1* increase the risk for a new tumor predisposition syndrome characterized by several tumors, including uveal melanoma [41]. A high expression of DNA repair genes was found in primary melanomas with higher metastasis rates and lower chemotherapy sensitivity [42]. An interesting study described the following new mechanism for DNA damage response in melanoma: after DNA damage induced by cisplatin, the key HR protein RAD51 was inactivated and the activity of the translesion polymerase ζ was increased, allowing cells to proliferate in the presence of the chemotherapeutic agent [43].

All together, these data indicate that alterations in the HR pathway are frequent in melanoma. These HR defects can sensitize melanoma to PARP inhibitors.

## 3. Targeted Therapy and Immunotherapy in Melanoma

The prognosis of patients with melanoma has been improved by the recent development of novel therapeutic approaches, such as inhibitors of the MAPK pathway and immune checkpoints. Thanks to these treatments, the five-year overall survival (OS) for metastatic melanomas increased from less than 10% to 40% [21].

The fundamental role of the MAPK pathway in melanoma genesis has opened the road to the discovery of small molecular inhibitors targeting BRAFV600E (vemurafenib, dabrafenib, encorafenib) and the kinases MEK1/2 (trametinib, cobimetinib, binimetinib) (Figure 2A). Vemurafenib and dabrafenib have significantly improved response rates of about 50%, progression-free survival (PFS) and overall survival (OS), compared to chemotherapy in patients with metastatic melanomas carrying the BRAFV600E mutation [44,45]. However, the use of BRAF and MEK1/2 inhibitors is inevitably related to the acquisition of drug resistance, with most patients relapsing within one year. The combination of BRAF and MEK1/2 inhibitors has been shown to delay the onset of drug resistance, increasing the overall response rate (ORR), with respect to monotherapy with dabrafenib [46]. The median OS for metastatic melanoma patients treated with BRAF and MEK1/2 inhibitors is between 22 and 33 months and the 5-year OS rate is 34% [47,48]. It is now clear that the major obstacle to the long-term benefit of targeted therapies is the development of acquired resistance, even in the combination regimen.

The second breakthrough in melanoma treatment was the discovery of immune checkpoint inhibitors (ICI) (Figure 2B,C). The first approved ICI was ipilimumab, a monoclonal antibody anti-cytotoxic T lymphocyte-associated protein 4 (CTLA-4), which increased survival over the chemotherapeutic agent dacarbazine [49]. Subsequently, the anti-programmed death 1 (PD-1) monoclonal antibodies nivolumab and pembrolizumab demonstrated increased efficacy in comparison to ipilimumab, with less toxicity. These monoclonal antibodies accomplished an overall response rate (ORR) of 40 to 50% and a five-year OS rate of 41–44% [50,51,52,53]. Further clinical trials showed enhanced efficacy of nivolumab and ipilimumab in combination, compared to monotherapy [52,53]. The major shortcomings of immunotherapy are the lack of predicted biomarkers of therapy outcome and toxicity, which can be difficult in some patients, limiting the duration of the treatment. Therefore, novel therapeutic approaches are needed for patients who are resistant or do not respond to the available targeted therapy and immune checkpoint inhibitors.

## 4. DDR Inhibitors in Melanoma

Experimental evidence suggests that deregulated DDR may contribute to melanoma aggressiveness and survival; thus, DDR can be therapeutically exploited in this tumor type. In this paper, we will depict the current evidence on the effect of PARP, ATM, CHK1, WEE1 and ATR inhibitors both in preclinical and clinical settings in melanoma (Figure 1), both as monotherapy and in combination with other targets belonging to the pathways crucial for melanoma proliferation and survival. Moreover, since recent studies have shed light on the immunomodulatory role of the DDR targets, we will overview the recent data on the crosstalk between DDR inhibitors and immune checkpoint inhibitors.

### 4.1. PARP Inhibitors in Melanoma

PARP inhibitors (PARPi) act through the following different mechanisms: the first is the inhibition of the canonical PARP function, the second is PARP “trapping”, in which activated PARPs are blocked (trapped) on DNA lesions and block the progression of replication forks, causing the accumulation of single strand breaks, increasing genomic instability and cell death [54]. The last mechanism suggests that PARPi toxicity is due to replication gaps [55]. The cells with HR defects, such as alterations in *BRCA1/2* or other genes, are unable to repair DSBs induced by PARPi and undergo death with synthetic lethality [56,57]. PARPi are currently used in cancer therapy to treat patients with HR deficiencies or *BRCA* mutant tumors, such as breast, ovarian, pancreatic and prostate cancers [58,59,60,61].

Several in vitro studies reported that PARPi reduce migration and invasion and induce apoptosis in different melanoma cell lines, even in the absence of HR alterations [62,63,64] (Table 1). It is noteworthy that treatment with veliparib showed a robust effect in a melanoma cell line resistant to dabrafenib, prospecting a potential therapeutic option for resistant melanomas [62]. An interesting study reported the effect of PARPi on PDX of melanoma cells with the following differential HR statuses: MM425X (*BRCA1* and *ARIDB1* mutated), MM390X (*CHD2* mutated) and MM507X (wild type). The treatment with niraparib resulted in a decrease in cell viability and induction of apoptosis in both mutated cell lines, but had a slight effect on the wild type one. A similar effect, accompanied by a reduction in cell invasion, was obtained in the established human melanoma cell lines LOX (*ARID1A* and *BAP1* mutated). The effect of PARP inhibition was also investigated on MM425X and MM507X in NSG mice. The daily administration of niraparib (25 mg/kg) and olaparib (50 mg/kg) resulted in a significant anti-tumor effect in both xenografts [34].

On the basis of the preclinical data obtained in vitro and in vivo, PARPi could be considered as an option for melanoma treatment, in particular in patients with HR defects. In clinical trials, the inhibition of PARP was studied in melanoma in combination with chemotherapy, in particular temozolomide, in order to overcome chemoresistance. Two trials evaluated the therapeutic outcome of temozolomide combined with veliparib [86] or rucaparib [87] in patients with advanced metastatic melanoma. Both studies showed an improvement in PFS, without reaching a statistical significance. These unsatisfying results may be due to the fact that patients were not stratified according to HR status. Indeed, a recent case report showed the partial response of olaparib as monotherapy in a metastatic melanoma patient carrying a *PALB2* mutation, who has progressed in immunotherapy [88]. This study confirmed the importance of the evaluation of HR status in melanomas to a better therapeutic choice. Currently, there are three ongoing studies using niraparib, which are as follows: the first is assessing its efficacy in a cohort of metastatic melanomas with generic HR mutations (NCT03925350), the second is investigating its effect in different solid tumors, including uveal melanoma, carrying *BAP1* and other DNA damage repair (DDR) mutations (NCT03207347) and the last one is studying different solid tumors, including melanoma with pathogenic or likely pathogenic *PALB2* mutations (NCT05169437).

A further therapeutic strategy for the use of PARPi in melanoma could be the combination with other kinds of therapies. Weigert and colleagues suggested that PARPi could increase the effect of radiotherapy, inducing G2/M arrest leading to cell death [67]. Otherwise, PARPi could also be used in combination with immunotherapy. Melanoma is highly sensitive to immune system activation and for this reason is refractory to immunotherapy [89]. A recent case report described the near-complete response of a metastatic melanoma patient with a high homologous recombination deficiency (HRD) score after treatment with nivolumab in combination with olaparib, after progression on prior nivolumab monotherapy. The patient also showed complete mutation clearance after treatment [90]. To date, several clinical trials are ongoing to evaluate the efficacy of PARPi and immunotherapy in melanoma with homologous recombination defects (NCT04633902, NCT04187833). The active clinical trials with PARP inhibitors and the other DDR inhibitors herein overviewed are summarized in Table 2.

### 4.2. ATM Inhibitors in Melanoma

ATM plays a pivotal role in DSB repair, in particular in the HR process. ATM seems to be implicated in melanoma susceptibility. A large multicenter study demonstrated that ATM loss of function variants have a higher frequency in melanoma, compared to healthy individuals from a large multicenter melanoma cohort [91]. In addition, Bhandaru and colleagues suggested a possible correlation between the expression of phosphorylated ATM (Ser 1981), melanoma progression and patient survival [92]. ATM inhibition increases genomic instability and, consequently, the sensitivity of cancer cells to radiotherapy. In uveal melanoma, cellular radioresistance seems to be correlated with ATM protein levels. The treatment with ATM inhibitor AZD1390 enhanced cell sensitivity to both X-ray and proton irradiation [68]. Currently, there are at least three ATM inhibitors (AZD0156, KU-60019, AZD1390) undergoing clinical trials in solid tumors [93], but so far, none of these were investigated in melanoma, neither as a single agent nor in a combination regimen.

### 4.3. CHK1 Inhibitors in Melanoma

CHK1 is a key regulator of the DDR pathway. It is required in the S phase of the unperturbed cell cycle to regulate origin firing, to circumvent DNA breakage, thus, maintaining cancer cell survival under replication stress [4,94,95]. CHK1 also plays a crucial role in the G2 phase, controlling the correct mitotic entry [12,96,97]. High expression levels of CHK1 are correlated with worse prognosis in melanoma [73]. Melanomas, including the metastatic counterpart, are characterized by high levels of endogenous replicative stress [98], thus, particularly relying on the activity of CHK1 to maintain the DNA damage under a tolerable level for their survival. Preclinical studies showed that melanoma cells with high levels of endogenous replicative stress are particularly susceptible to CHK1 inhibitors (AR323, AR678, GNE-323 and GDC-0575) in vitro and in vivo in xenograft models [74,76]. The concept at the basis of this sensitivity is that CHK1 inhibition treatment enhances this replication stress, due to the inability of homologous recombination repair to fix the DSB, as this process requires CHK1 function [95]. The in vivo hypoxic conditions featuring melanoma were also shown to predispose a CHK1 inhibitor response. CHK1, and more in general the DDR pathway activation, are triggered by low oxygen levels [99,100]. This process was shown to be associated with hypoxia-inducible transcription factor 1a (HIF-1a), the main transcription factor involved in the control of cellular adaptation to low oxygen levels. The inhibition of HIF-1a degradation by dimethyloxalylglycine (DMOG) has been shown to activate CHK1 and to sensitize melanoma cells to the CHK1/2 inhibitor AZD7762 in vitro [70]. Moreover, the increased vulnerability of melanoma cells to CHK1 inhibitors, under amplified hypoxic conditions, could be pharmacologically exploited in vivo. In their study, Possik and colleagues demonstrated that bevacizumab, which neutralizes VEGF (increasing in in vivo hypoxic areas), is a valuable synthetic lethal partner of CHK1 inhibition in melanoma [70]. Importantly, under hypoxic conditions, AZD7762 kills cells both sensitive and resistant to BRAF inhibition and induces a significant tumor growth delay in vivo in PDX melanoma cells with primary resistance to vemurafenib (PLX4032) [70]. Other evidence recently confirmed the advantage of the use of CHK1 inhibitors in melanoma cells both sensitive and resistant to BRAF inhibitors. A recent study showed that the CHK1 inhibitor PF-477736 can act synergistically with PLX4032 to inhibit cell growth in PLX4032-resistant melanoma cells and mouse xenografts. [73]. Again, CHK1 inhibitors (specifically the compounds AZD-7762 and CHIR124), together with the WEE1 inhibitor adavosertib, were among the kinase inhibitors displaying the best activity in combination with BRAF inhibitors (dabrafenib and vemurafenib) in a kinase inhibitor library screening, recently conducted in BRAF mutated melanoma cell lines (sensitive or made resistant to BRAF inhibitors) [71]. An additional example of success in combining CHK1 inhibitors with inhibitors of targets involved in the pro-survival pathway in melanoma was given by the observed synergistic effect in combining the AXL inhibitor BGB324 with the CHK1 inhibitor AZD-7762 [72]. The receptor tyrosine kinase AXL has been found overexpressed in a wide range of cancers, including melanoma, and it was shown to mediate resistance to both target therapy (BRAF and MEK inhibitors) and immunotherapy [101,102]. The drug combination inhibited cell proliferation and tumor growth and induced apoptosis. The concept that hampering pro-survival pathways may enhance the activity of CHK1 inhibitors in specific cellular contexts has also been demonstrated recently in other tumor types [103,104,105]. Combining CHK1 inhibitors with other DDR target inhibitors is also an emergent approach, widely demonstrated in preclinical models of many cancer types [106,107,108,109]. The synergistic effect of the combined CHK1 inhibitors with the WEE1 inhibitor adavosertib has been demonstrated in malignant melanoma [69]. The combined treatment reduced cell proliferation and viability, spheroid growth and inhibited tumor growth in melanoma xenografts [69]. Interestingly, the combined treatment was strikingly effective, both in the BRAF inhibitor sensitive and in primary and acquired BRAF inhibitor resistant melanoma cell lines [71]. This combined treatment has a much lower effect in normal cell lines (normal fibroblasts and melanocytes), suggesting a low risk of toxicity in vivo, thus, warranting further clinical development [69,107].

Several initial preclinical studies in many cancer types, including melanoma, showed that CHK1 inhibitors act to potentiate chemotherapy, abrogating the G-M checkpoint, with a stronger activity in a p53 deficient tumor setting [12,78,110]. Unfortunately, the development of CHK1 inhibitors in clinical trials has a long and somehow disappointing history. The majority of phase I/II clinical trials with CHK1 inhibitors has been conducted in combination with chemotherapy in advanced solid tumors (including malignant melanoma) and hematologic malignancies (prevalently with antimetabolites) [111,112], but none of these have yet reached phase III evaluation or FDA approval [113], due to the development of adverse side effects, especially when used in a combination regimen. Recently, a synergism of the CHK1 inhibitor GDC-0575, in combination with low concentrations of the reversible ribonucleotide reductase inhibitor hydroxyurea (HU), was observed in melanoma cells [75]. This drug combination was shown to be preferentially effective in killing melanoma cells while safeguarding healthy cells, since low doses of HU only reversibly activate CHK1 through the DNA-PK pathway [114]. On the other hand, normal cells treated with the CHK1 inhibitor, in combination with submicromolar concentrations of gemcitabine, completely lost proliferative potential, due to the irreversible ATR–CHK1 pathway activation after gemcitabine treatment [75].

Currently, only the CHK1 inhibitor prexasertib (LY2606368) has active, ongoing clinical trials. In phase I trials, prexasertib in monotherapy has been shown to be relatively well tolerated, with only transient and reversible low side effects [115,116]. Promising phase 2 studies have been conducted in monotherapy in high-grade serous ovarian cancer [117] and in combination with standard therapy or the PARP inhibitor Olaparib [118,119]. However, recently, results from a phase I study of the combination of prexasertib with ralimetinib (p38 mitogen-activated protein kinase inhibitor) in colorectal or non-small cell lung cancer mutated in KRAS and/or BRAF failed to achieve the escalation to phase II [120]. At the moment, a phase II study of prexasertib in patients with solid tumors with high replicative stress or homologous recombination deficiency (both features also characterize melanoma) is active and results are expected soon (NCT02873975).

CHK1 has a role in the innate immune response to genotoxic stress. Depending on the cancer background setting, it has been shown that CHK1 inhibitors may either upregulate PD-L1 expression and then increase the response to anti-PD-L1 [121], or decrease PD-L1 expression after DSB [122]. Prexasertib, combined with the anti-PD-L1 LY3300054 antibody, enhances anti-tumor T cell activation in patients with high grade serous ovarian tumors and other solid tumors [123]. In the context of melanoma, a recent study showed that targeting replication stress, using a CHK1 inhibitor in combination with low doses of HU, led to increased DNA damage that can activate the cGAS-STING pathway, thus, inducing a pro-inflammatory cytokine and chemokine expression. This, in turn, can trigger an anti-tumor immune response and ultimately promote immunogenic cell death. These observations were corroborated by two different CHK1 inhibitors (GDC-575 and SRA737), suggesting that the effects are specifically due to CHK1 inhibition [77]. CHK1 inhibition combined with HU induced DNA damage and promoted an increased expression of PD-L1 on tumor cells [75]. In addition, this combination induced a tumor-associated immune cell population (including Lag3, TIM3 and NKT), which in turn strongly upregulates PD-L1. Anti-PD-1, in combination with CHK1 inhibitors + HU, did not produce a potentiating effect, suggesting that this treatment (CHK1 inhibitor + HU) activates more prevalent mechanisms maintaining an immunosuppressive microenvironment and does not negatively affect immune responses, triggering a strong anti-tumor immune response [77].

### 4.4. WEE1 Inhibitors in Melanoma

WEE1 works as a key gatekeeper of mitotic entry and it controls DNA replication, moderating the firing of replication origins, promoting homologous recombination, and blocking the inappropriate resection of stalled replication forks [13]. WEE1 inhibition enhances replicative stress and DNA damage, and leads to cell cycle dysregulation [4,13,14,124]. WEE1 is upregulated in melanoma compared to benign nevi and high WEE1 expression indeed correlated with a poor prognosis in melanoma patients [125]. Its inhibition by siRNA increased DNA damage and cell death in melanoma cell lines, regardless of p53 status [125]. However, the role of WEE1 expression in melanoma progression and aggressiveness is somehow controversial. The administration of miR-195 targeting WEE,1 in combination with doxorubicin, significantly reduced G2-M cell cycle arrest, which was re-established by stable overexpression of WEE1. On the other hand, miR-195 administration increased melanoma proliferation and its overexpression enhanced migration and invasiveness of melanoma cells, suggesting a role for this kinase in inhibiting migratory signaling [126]. Conversely, miR-155 expression, which regulates the expression of WEE1, was shown to be lost in patients who developed metastatic melanoma. The inhibition of WEE1 by either overexpression of miR-155 or siRNA results in a significant decrease in metastasis in a mouse model of melanoma [127]. A recent study showed the successful use of a new siRNA delivery system with nanoparticles displaying highly penetrating abilities. They were recently used as a carrier to deliver WEE1 siRNA in melanoma preclinical models [128]. The nanoparticle/siWEE1 complex displayed a strong anti-cancer activity in vitro in B16 cells and an anti-tumor effect in vivo in subcutaneous xenograft and lung metastasis of B16 tumor models, as compared with the negative control group [128].

Similar to the CHK1 inhibitors, WEE1 inhibitors were also primarily shown to abrogate the G2-M checkpoint in many tumor types, including melanoma [78,79]. Adavosertib (previously named AZD-1775) is a pyrazolo-pyrimidine derivative, acting as an ATP competitive potent and selective small molecule inhibitor of WEE1, and displays cytotoxic activity as a single agent in many cancer cell lines, including melanoma cells [83,129,130]. Adavosertub enhanced the cytotoxic activity of different DNA damaging agents both in vitro and in vivo, including antimetabolites, DNA crosslinking agents and topoisomerase inhibitors, with a higher activity observed in p53-deficient/mutant experimental models [129,131].

As mentioned above, adavosertib was observed in a recent kinase screening among the most effective kinase inhibitors to work synergistically with BRAF inhibitors [71]. Moreover, as already depicted in the CHK1 inhibitors section, synthetic lethality and therapeutic synergy, combining adavosertib with CHK1 inhibitors and with other DDR inhibitors, has been further successfully demonstrated in many tumor types, including melanoma [69,78,109,132]. Interestingly, a recent study showed that metastatic uveal melanoma with high MYC activity is highly susceptible to the WEE1 inhibitor Adavosertib [84].

Interestingly, WEE1 kinase is a downstream target of BRAFV600E [133]. A recent study identified WEE1 as a valid target to inhibit in combination with AKT3, a major target in melanoma. The combined inhibition of AKT3 and WEE1 kinases synergistically inhibited cellular proliferation and induced apoptosis in melanoma cells [80].

Currently, the efficacy of adavosertib is being evaluated in phase I/II clinical trials for the treatment of advanced solid tumors. A phase II trial of adavosertib in combination with olaparib is ongoing in patients with tumors harboring *TP53* and/or *KRAS* mutations (NCT02576444). Adavosertib is among the targeted drugs included in the MATCH screening trial (NCT02465060) currently ongoing in patients with solid tumors (including melanoma) or lymphomas that have progressed following at least one line of standard treatment. Specific genetic tests are being conducted to detect the patients’ genetic abnormalities (such as mutations, amplifications, or translocations) and to identify the specific target drug that is possibly effective for the specific genetic lesion. A phase I trial is ongoing to assess the efficacy and tolerability of adavosertib with MEDI4736 (durvalumab), a monoclonal antibody against PD-L1, in refractory solid tumors (NCT02617277). A recent study showed that WEE1 inhibition is not able to induce immunogenic cell death or to enhance PD-L1 expression in tumor cells, but is able to synergize with radioimmunotherapy in melanoma xenografts models [82].

Although the phase I/II trials with adavosertib have been ongoing for more than 10 years, there are no phase III trials yet. A current effort is being undertaken to develop new and more selective WEE1 inhibitors, since it has to be taken into account that adavosertib is also known to inhibit, although at a lesser extent, other kinases (e.g., PLK1) [134]. Recently, a highly selective and potent WEE1 inhibitor has been identified named ZN-c3 (by Zentalis Pharmaceuticals), which showed excellent in vivo efficacy [135]. It is currently in phase I/II clinical trials, evaluating the efficacy and tolerability in patients with solid tumors (including malignant melanoma), both as a single agent and in combination with other drugs (NCT04158336).

### 4.5. ATR Inhibitors in Melanoma

ATR is one of the main upstream regulators of the DDR pathway. It leads to cell cycle arrest, DNA repair, and to the suitable control of stalled replication forks, through the propagation of the DNA damage response signal to CHK1, its major kinase target [4,19]. Preclinical data suggested that the synthetic lethal targeting of ATR in tumors with high oncogenic replicative stress, or in tumors reliant on the ATR pathway (such as the loss of *TP53*, *ARID1A*, or *ATM*), may represent a valid therapeutic advantage [136,137,138]. It is conceivable that melanoma, characterized by high levels of replication stress, may be highly responsive to ATR inhibitors. The relatively recent release of the ATR crystal structure took the route for the development of ATR inhibitors [139]. The evidence from recent studies suggest that ATR inhibition may be compensated by parallel pathways that activate CHK1 independently from ATR [114,140] and the relatively low levels of ATR in unperturbed cells [114,141] pushed the quick development of ATR inhibitors, showing to be safe and not toxic for healthy cells [141,142]. At the moment, the most advanced ATR inhibitors have completed phase I as single agents. The ATR inhibitors showing clinical efficacy are ceralasertib (AZD6738), berzosertib (VX-970/M6620), and elimusertib (BAY1895344) [142,143,144].

Interestingly, a recent work showed that BRAFV600 mutant cell lines, with primary or acquired in vitro resistance to BRAF and MEK inhibitors, are highly susceptible to the combined treatment with the ATR inhibitor AZD-6738 and the PARP inhibitor olaparib [65]. This observed synergistic effect once again highlights the putative significant therapeutic potential of the DDR-DDR inhibitor combinations in melanoma. Recent preclinical evidence showed that the therapeutic combination of ATR and BET protein inhibitors is effective in melanoma, corroborating the same effects observed previously in MYC-induced lymphoma [85].

The major evidence attributing a therapeutic value in melanoma to ATR inhibitors comes from recent clinical trial results. Recently, a phase I study of ceralasertib, combined with paclitaxel administered weekly in refractory advanced solid tumors, also enrolled 33 patients with melanoma resistance to treatment with anti-PD1 therapy (NCT02630199). In the melanoma cohort, 11 patients with metastatic melanoma, who were previously resistant to PD-1 inhibitors, reached long-lasting responses [145,146]. Interestingly, these responses after progression on an ICI were clearly observed in melanoma subtypes with different histopathologic and mutational profiles (cutaneous, acral, and mucosal melanoma) [146].

In addition, a phase II study (NCT03780608) of ceralasertib in combination with durvalumab was assessed in patients with metastatic melanoma, who were not responding to anti-PD-1 therapy. Thirty metastatic melanoma patients, previously treated with anti-PD-1, were enrolled (twenty-three primary resistant). The ORR among the evaluable patients was 30% (9 out of 30 patients) and the disease control rate (DCR) was 63.3% (19 out of 30 patients). The response to the treatment combination did not correlate with previous immune checkpoint inhibitor responses. Interestingly, tumors with immune-enriched microenvironments or alterations in the DDR pathway had major benefits [147].

Taken together, these data pointed to the emerging role of ATR in the tumor immune microenvironment and suggested a significant benefit of using ATR inhibitors, in combination with immunotherapy, in melanoma. In support of this evidence, a recent analysis from The Cancer Genome Atlas and The Cancer Immunome Atlas showed that samples mutated in DNA damage response genes, including *ATR*, present high neoantigen levels [148].

Recently, a work from Chen and colleagues showed that *ATR* mutations modulate the tumor immune microenvironment in melanoma models, leading the immune system to accelerate tumor growth. Homozygous *ATR* mutated melanomas showed a reduction in the number of infiltrating CD3+ T cells, but an increase in the infiltrating macrophages and B cells, compared to *ATR* wt or hemizygous mutated tumors. This *ATR* deficient status correlated with an increased neo antigen expression (including PD-L1) that suppressed the immune response, facilitating tumor growth [149]. Similar to CHK1 inhibition, recent studies showed that the pharmacological inhibition of ATR may induce cGAS-STING-mediated anti-tumor immunity and may trigger tumors for immune checkpoint blockade [150,151]. Mechanistically, the cytosolic DNA fragments, derived from unrepaired DNA damage induced by ATR inhibition, interact and lead to the activation of cGAS-STING signaling [152]. Other recent studies have demonstrated the synergistic contribution of ATR inhibitors with radioimmunotherapy [153,154].

## 5. Conclusions

The biological complexity of melanoma and the plethora of mutations responsible for its development and aggressiveness, together with the rapid development of its resistance to the current available targeted and immune therapies, suggest a need to devise more effective therapeutic strategies. The preclinical and clinical evidence herein overviewed showed that the use of DDR inhibitors in melanoma may have the potential to both efficiently synergize with the current therapies and to also overcome the resistance to such therapies (Figure 3). More efforts should be undertaken, especially in clinical trials, to rationally exploit DDR inhibitors in a tailored manner, taking advantage of the synthetic lethal interactions observed in preclinical studies (e.g., HR defects and high endogenous replicative stress) and to smartly combine them (e.g., with target modulators of pro-survival pathways or with other DDR inhibitors) to hit at the same time the different pathways on which melanoma relies on.

## Figures and Tables

**Figure 1 cells-11-01466-f001:**
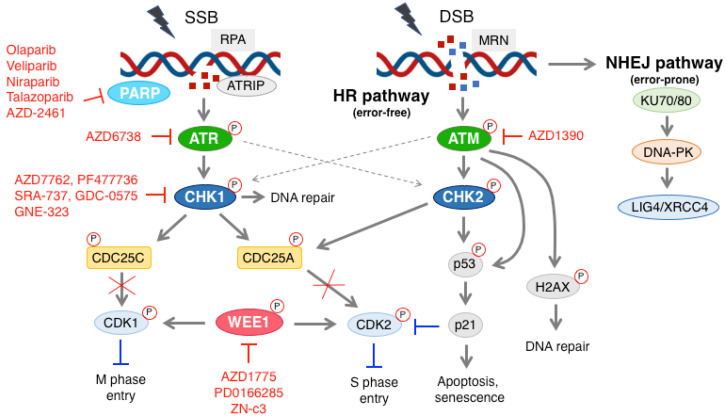
Schematic representation of the DDR pathway and DDR inhibitors used in preclinical and clinical settings in melanoma. Single strand break (SSB) is recognized by the RPA complex, which leads to the activation of ATR and CHK1. The latter promotes phosphorylation and inactivation of the phosphatases CDC25A and CDC25C, which are involved in dephosphorylation and activation of CDK2 and CDK1, respectively. CHK1 may also be activated by ATM. WEE1 regulates the CDKs activity in a negative manner, playing an essential role in controlling S and M phase entry. The MRN complex recognizes double strand break (DSB), and activates ATM. CHK2 and p53 are the main targets of ATM. Upon activation, CHK2 and p53 can lead to either cell cycle block or apoptosis. Active ATM can also phosphorylate H2AX, which can lead to DNA repair. During SSB formation, ATR can also activate CHK2. PARP, ATR, CHK1, ATM and WEE1 inhibitors are indicated in red. More information about these inhibitors is reported in the main text. DSB can also activate the NHEJ pathway, whose main components include KU70/80, DNA-PK and LIG4/XRCC4. P with red circles stands for phosphorylation.

**Figure 2 cells-11-01466-f002:**
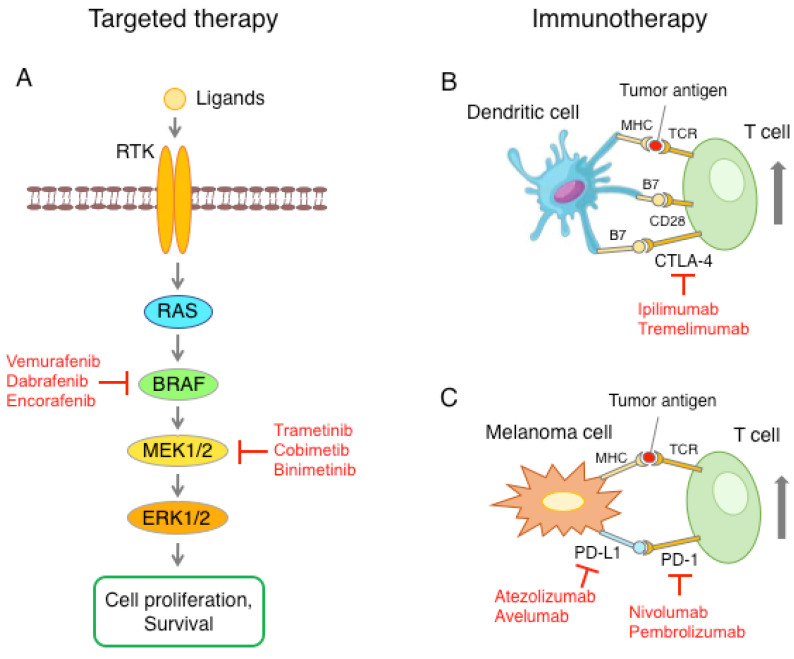
Schematic representation of the main therapeutic approaches against melanoma. (**A**) The RAS-RAF-MEK1/2-ERK1/2 pathway is often upregulated in melanoma, with a consequent increase in cell proliferation and survival. Pharmacological inhibition of mutant BRAF or kinase MEK1/2 blocks the MAPK pathway. (**B**) Activation of T cells requires presentation of the tumor antigen to the T-cell receptor (TCR) through the major histocompatibility complex (MHC) and the interaction of CD28 with B7 ligand, present on the surface of dendritic cells. T cells also express CTLA-4, whose binding to B7 triggers the signal to inactivate T cells. Anti-CTLA-4 antibodies inhibit the interaction between CTLA-4 and B7, leading to T cell activation. (**C**) Melanoma cells can express high levels of PD-L1, which, through the interaction with the PD-1 receptor on the T cells, causes a reduction in T cell function. Blockade of the PD-L1/PD-1 axis with anti-PD-1 or anti-PD-L1 checkpoint inhibitors restores the ability of T cells to recognize and destroy melanoma cells.

**Figure 3 cells-11-01466-f003:**
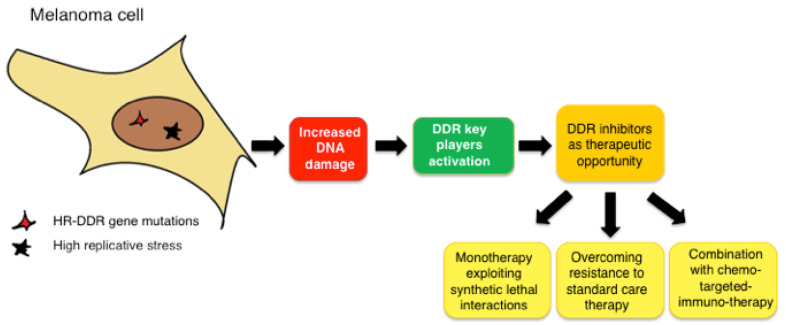
Schematic representation of the therapeutic advantage of using DDR inhibitors in melanoma.

**Table 1 cells-11-01466-t001:** DDR inhibitors with preclinical proof of cytotoxic and anti-tumor effects in melanoma.

Target	Compound Name	Reference
PARP	Veliparib	[62]
	Olaparib	[34,64,65,66]
	Niraparib	[34,67]
	Talazoparib	[67]
	AZD2461	[63]
ATM	AZD1390	[68]
CHK1	AZD-7762	[69,70,71,72]
	PF-477736	[73]
	AR-323, AR-678	[74]
	GDC-0575	[75,76,77]
	GNE-323	[76]
	SRA-737	[77]
	CHIR-124	[71,78]
WEE1	PD0166285	[79]
	WEE1 inhibitor II	[78]
	Adavosertib (AZD-1775)	[69,71,80,81,82,83,84]
ATR	Ceralasertib (AZD-6738)	[65,85]

**Table 2 cells-11-01466-t002:** DDR inhibitors currently undergoing clinical trials in melanoma.

DDR Target	Agents	Phase	Trial ID *
PARP	Niraparib	II	NCT03925350
	Niraparib	II	NCT05169437
	Niraparib	II	NCT03207347
	Olaparib + Pembrolizumab	II	NCT04633902
	Talazoparib + Nivolumab	II	NCT04187833
	Veliparib + Paclitaxel + Carboplatin	I	NCT01366144
WEE1	Adavosertib ZN-c3	II I	NCT02465060 NCT04158336
ATR	Ceralasertib + paclitaxel	I	NCT02630199
	Ceralasertib + durvalumab	II	NCT03780608

* Recruiting and active, not recruiting clinical trials specifically involving melanoma were included (status on Clinicaltrial.gov, accessed on 8 March 2022).

## Data Availability

Not applicable.

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
