# Peer review of "Novel Therapeutic Approaches with DNA Damage Response Inhibitors for Melanoma Treatment"

_cells, 2022, doi:10.3390/cells11091466_

Round 1

Reviewer 1 Report

To my opinion, it's written very well in a comprehensive manner.
It touches the main DDR pathways and it's molecular components and is discussing the the use of DDR-inhibitors, also in combination with other therapy options in the field of melanoma treatment. It gives a good overview about clinical trial going on in the moment.
This is an important summary, helpful for experts and newcomers in the field. Literature references are chosen well and in many cases very recent.

Author Response

We thank the Reviewer for the very positive comments on our manuscript.

Reviewer 2 Report

The review article presented by Maresca and collaborators is timely and is well written, illustrated and referenced in the literature. Personally, I believe it is ready for publication , but I will make a small suggestion in order to broaden the audience of interest and increase the chances of citation. In this review, the role of DNA damage response in the biology and therapeutic response in melanoma is presented. The strengths of the review are the molecular details provided in the description of cellular events and signaling pathways. The weak point is the specificity of the topic addressed. To circumvent this limitation and increase the chances of citation of the review, in the item “Targeted therapy and immunotherapy in melanoma” line 155, the authors could add a Figure containing information on the interaction of immune system checkpoints and the drugs/strategies used to inhibit them. I believe that this information could please readers and broaden the audience of interest.

Author Response

We thank the Reviewer for the very positive comments on our manuscript. As suggested by the reviewer, we added a new Figure (new Figure 2) reporting a schematic representation of the most important therapeutic approaches to melanoma treatment, including targeted therapy and immunotherapy.

Reviewer 3 Report

The manuscript is well designed about melanoma treatment modalities that need emergency for the successful cure of patients. Figure 1 and Figure 2 are well designed but maybe it is better to polish their presentation style with colours and arrows which are adjusted in a design.

Author Response

We thank the Reviewer for the very positive comments on our manuscript. As suggested by the reviewer, we slightly changed colors and style in Figure 1.